# Anticancer Effects of Cold Atmospheric Plasma in Canine Osteosarcoma Cells

**DOI:** 10.3390/ijms21124556

**Published:** 2020-06-26

**Authors:** Jaehak Lee, Hyunjin Moon, Bonghye Ku, Keunho Lee, Cheol-Yong Hwang, Seung Joon Baek

**Affiliations:** 1College of Veterinary Medicine and Research Institute for Veterinary Science, Seoul National University, Seoul 08826, Korea; ljh930307@snu.ac.kr (J.L.); hyjin1995@naver.com (H.M.); cyhwang@snu.ac.kr (C.-Y.H.); 2R&D Center, PSM Inc. Jungwon-gu, Seongnam-si, Gyeonggi-do 13207, Korea; kbh@ionmedical.co.kr (B.K.); kholee@ionmedical.co.kr (K.L.)

**Keywords:** canine osteosarcoma, cold atmospheric plasma, high-content screening, reactive oxygen species, metastasis

## Abstract

Osteosarcoma is known to be one of the frequently occurring cancers in dogs. Its prognosis is usually very poor, with a high incidence of lung metastasis. Although radiation therapy has become a major therapeutic choice for canine osteosarcoma, the high costs and unexpected side effects prevent some patients from considering this treatment. Cold atmospheric plasma (CAP) is an ionized gas with high energy at low temperatures, and it produces reactive oxygen species that mediate many signaling pathways. Although many researchers have used CAP as an anticancer therapeutic approach in humans, its importance has been neglected in veterinary medicine. In this study, D-17 and DSN canine osteosarcoma cell lines were treated with CAP to observe its anticancer activity. By high-content screening and flow cytometry, CAP-treated cells showed growth arrest and apoptosis induction. Moreover, the osteosarcoma cells exhibited reduced migration and invasion activity when treated with CAP. Overall, CAP exerted an anticancer effect on canine osteosarcoma cell lines. CAP may have the potential to be used as a novel modality for treating cancer in veterinary medicine.

## 1. Introduction

Cancer, or malignant neoplasia, is an array of diseases in which some cells display uncontrolled growth and metastasis. Cancer affects individuals of all age groups, including fetuses, but the risk tends to increase with age. According to the American Veterinary Medical Association, approximately one in four dogs will, at some stage of their life, develop neoplasia. The rate is similar to that in humans, where one in three individuals develop cancer at some point in their lifetime. Since canine and human metabolic pathways and drug responses differ, there is a pressing need for new treatments regarding canine cancer.

Canine osteosarcoma is common, especially in large breeds. The incidence rate of osteosarcoma is 27 times higher in dogs than in humans [1]. The prognosis is poor, with 1-year and 2-year survival rates of 43.2% and 13.9%, respectively, with pulmonary metastasis [2]. There are various treatments for osteosarcoma, including amputation, limb-sparing surgery, chemotherapy and immunotherapy, but stereotactic radiotherapy (SR) has recently become a popular treatment approach for canine osteosarcoma in the United States. SR could be an alternative for dogs that are expected to have poor surgical outcomes, or whose owners refuse amputation [3]. In addition, one study showed that among 46 dogs treated with SR, approximately 63% developed fractures, which may have resulted from the increased activity levels following the analgesia caused by the radiotherapy [4].

Plasma medicine is an emerging field that has proven beneficial for hospital hygiene, antifungal treatment, dental care, skin diseases, chronic wounds and cosmetics [5]. Previous efforts focused on detecting and identifying the basic processes of plasma had an aim of designing plasma machines for specific applications. Although most studies focused on dielectric-barrier discharge (DBD) methods, we recently reported the use of microwave plasma to induce immunogenic cell death in human cancer cells [6]. Cold atmospheric plasma (CAP) is ionized gas with high energy at relatively low temperatures. CAP generates reactive oxygen species (ROS), which can have many biological effects, including wound healing, sterilization, gene transfection, blood coagulation and cancer therapy [7,8,9]. Plasma medicine recently became a new area in human medicine [10] due to its mechanisms—including DNA damage, apoptosis, necrosis or autophagy by stimulating many ROS-mediated signaling pathways—leading to anticancer activity [11,12,13]. Recent studies have demonstrated that plasma can induce immunogenic cell death in cancer cells, which provides a potential application in cancer immunotherapy [6,14,15]. The use of CAP for biomedical treatments is an exciting new application, and could take plasma further in systematic medical research.

Although CAP has been studied in human medicine for treating cancer, it has not been applied in veterinary medicine. In this study, we demonstrated the anticancer effects of CAP on canine osteosarcoma cell lines, and proffered the possibility of using an innovative cancer treatment approach based on plasma energy. There is accumulating evidence that alternative cancer therapy, in combination with chemotherapy, is required to provide better results in cancer patients. Although our study focused on osteosarcoma, other canine cancers could be targeted using the same approach. This is the first report to elucidate the anti-cancer effects of CAP in canine cancer cell lines. This study may ground the possibility of using CAP in veterinary medicine to treat cancer.

## 2. Results

### 2.1. CAP Generation by Dielectric Barrier Discharge

CAP was generated by dielectric barrier discharge using helium gas, as previously described [12]. The device consists of electrodes and involves a dielectric mechanism, wherein the insulating material is placed between the electrodes and the aluminum oxide (Al_2_O_3_) layers. As shown in Figure 1A, the CAP device used in this study has an electrode thickness of 80–100 μm. For CAP generation, we used a 25-kHz alternating current power supply, using an insulated gate bipolar transistor inverting circuit for high voltage of plasma discharge. Plasma discharge was generated with a breakdown voltage of 450 V. CAP was applied to the canine osteosarcoma cell lines, and the anticancer effect was examined. The cells were cultured on 96-, 24- or 12-well plates; however, the distance from the nozzle to the surface of the media was consistently controlled to be 10 mm (Figure 1B,C). The electric power was 300 W, and the helium gas flow was set to 20 L per minute. As cold atmospheric plasma may generate unwanted heat, we examined the temperature after CAP treatment and found that the temperature of the media did not change even after 60 s (Figure 1D).

### 2.2. CAP Induces Cell Growth Arrest in Canine Osteosarcoma Cell Lines

To determine whether CAP modulated cell growth, two different canine osteosarcoma cells were treated with CAP and examined. CAP-treated cells exhibited a decreased cell count, as shown in Figure 2A. CAP had a greater effect on DSN cells, as CAP-treated DSN cells showed smaller cytoplasms than the CAP-treated D-17 cells. To evaluate the effect of CAP on cell proliferation, MTS assays were conducted. As shown in Figure 2B, both cell lines showed decreased cell viability in a time-dependent manner. We chose to focus on D-17 rather than on DSN because D-17 was more resistant to CAP. We also examined the cell growth change by staining cells with 5-ethynyl-2′-deoxyuridine (EdU), as this can be incorporated into the newly synthesized DNA. The fluorescence intensity was quantified by high-content screening (HCS), and the result revealed that longer treatments with CAP resulted in less de novo DNA synthesis (Figure 2C).

### 2.3. CAP Generates ROS and Causes DNA Damage in A Canine Osteosarcoma Cell Line

CAP can generate a cocktail of chemical agents, including ROS and reactive nitrogen species [16]. To examine whether CAP generated ROS in this study, D-17 cells were treated with CAP and then stained with 2′,7′-dichlorodihydrofluorescein diacetate (H_2_DCFDA), an indicator for ROS in cells. As shown in Figure 3A, the no-treatment (control) and gas-treated groups did not show any positive cells after staining, whereas green fluorescence was detected in the CAP-treated group, exhibiting ROS formation. This effect appeared in a time-dependent manner. To further quantify the fluorescence intensity, CAP-treated cells were analyzed with HCS technology after H_2_DCFDA and Hoechst 33342 staining. There was no significant difference between the control and gas-treated groups; however, ROS generation was observed in a time-dependent manner in the plasma-treated samples (Figure 3B). It has been known that ROS can cause DNA damage and induce ROS-mediated signaling pathways [17,18]. We tested whether DNA damage occurred after CAP was applied to D-17 cells. As expected, CAP-treated cells showed increased phospho-Histone-H2A.X (γH2A.X) levels, a marker for DNA damage (Figure 3C). The levels of phospho-Histone-H2A.X expression increased in a time-dependent manner. These results suggest that CAP generated ROS in a time-dependent manner, which resulted in DNA damage.

### 2.4. CAP Induces Apoptosis in a Canine Osteosarcoma Cell Line

It is known that DNA damage induces cell death [19,20]. Several assays were performed to determine whether CAP generates DNA damage in these cells. D-17 cells were exposed to CAP, stained with 4′,6-diamidino-2-phenylindole (DAPI), and then observed under a fluorescence microscope. Unlike the control or gas-treated groups, the cells treated with CAP showed nuclear condensation with a strong fluorescence intensity, which is an indicator for cell death (Figure 4A). The mitochondrial membrane potential, another marker for apoptosis, decreased in a dose-dependent manner when plasma was applied (Figure 4B). To confirm whether the CAP-induced cell death is apoptosis, we measured the apoptotic cells using flow cytometry. As shown in Figure 4C, Annexin V-positive cells increased in a time-dependent manner when treated with CAP. Altogether, these results imply that CAP can induce apoptosis in a canine osteosarcoma cell line.

### 2.5. CAP-Treated Canine Osteosarcoma Cells Showed Decreased Invasion and Migration Activity

A major concern in treating canine osteosarcoma is pulmonary metastasis [1]. Thus, we examined whether CAP may have the potential to suppress metastasis by analyzing cell invasion and migration activity. The cell migration assay was performed using scratching assay on a 24-well plate. When cells were grown to confluence, a scratch was made using a tip, and the cells were further grown for 48 h. As shown in Figure 5A, 70% of the area was refilled in the control group. However, CAP-treated cells filled with a gap of 55% and 30%, respectively, depending on the treatment time. As an additional method to measure metastasis, cell activity was measured by the three-dimensional tumor spheroid invasion assay. Spheroids from D-17 cells were constructed using an ultra-low attachment round bottom, and the basement membrane matrix was evaluated after CAP treatment. Both the control and gas-treated groups showed an increase in cell volume and invasion of the extracellular matrix, while the CAP-treated spheroids did not show any change in volume (Figure 5B, bottom graph) or invasion (Figure 5B, top panel). D-17 cells displayed outgrowth from the spheroids with invadopodia, whereas CAP-treated cells stayed as compact spheroids with a distinct border at the surrounding extracellular matrix. This indicates that plasma treatment results in the inhibition of the invasion activity of osteosarcoma cells.

## 3. Discussion

In this study, we treated canine osteosarcoma cell lines with cold atmospheric plasma (CAP) to test a new biomedical technique for treating cancer in veterinary medicine. Although studies investigating the therapeutic potential of CAP in humans are being conducted, little is known about this topic in veterinary medicine. Since human osteosarcoma is very rare compared to canine osteosarcoma [1], the higher incidence rate of canine osteosarcoma makes the dog companion population a good model for studying the human disease. Surgery with adjuvant chemotherapy [21], bisphosphonate therapy [22], radiotherapy [3] or immunotherapy [23] are the standard treatments in both human and canine patients, similar to how the metastasis is the common complication seen in human and canine osteosarcoma patients. 

Both D-17 and DSN showed decreased cell viability (Figure 2A,B), a similar result to that derived for human osteosarcoma cell lines [24]. CAP-treated cells exhibited decreased de novo DNA synthesis (Figure 2C). Since DNA replication happens during the cell cycle, we expect that the expression level of cyclins would decrease when treated with CAP. Indeed, CAP-treated cancer cells showed G2/M arrest with decreased cyclin B1 [25]. Similar results were observed in a range of different cell lines, including prostate, glioma, lymphoma, colorectal, and head and neck cancer [11], and this may imply that CAP can be applied not only to osteosarcoma, but to different types of cancer in veterinary medicine as well. Furthermore, the induction of cell proliferation with increased cyclin D1 was observed when the murine fibroblast cell line L929 was exposed to CAP [26]. This may support the feasibility of applying CAP to patients, in the sense that CAP may suppress cancer cell growth while promoting fibroblast proliferation, which would enhance wound healing after surgery.

The tumor microenvironment is a complicated and intricate condition, and is deeply involved in cancer development and metastasis. Particularly, interactions between hypoxia and reactive oxygen species (ROS) contribute to its complexity. This hypoxic condition leads to production of cellular ROS, followed by the induction of a ROS-dependent survival/proliferation pathway [27]. Interestingly, one of the genes expressed by ROS is the hypoxia-inducible factor (HIF), which induces angiogenesis-associated genes such as the vascular endothelial growth factor and other related genes [28]. In general, basal ROS levels in cancer cells are elevated, which may promote cancer growth. On the other hand, when ROS concentration is elevated beyond the threshold, tumor suppression, including apoptosis and autophagy, occurs. As shown in Figure 3, we demonstrated that CAP increases ROS production in osteosarcoma cells, leading to DNA breaks observed by detecting the up-regulation of the phosphorylated form of histone-H2A.X (γH2A.X), followed by the induction of apoptosis (Figure 4). Our results support the concept that increasing ROS levels would be beneficial to cancer patients. Currently, many anti-cancer drugs target the up-regulation of ROS to toxic levels [29]. 

Although ROS is a major biological component in plasma, other factors, including electromagnetic fields and ozone, play a role in anti-tumorigenesis. For example, ozone may be a good CAP component for consideration in treating cancer, because it is an activator of antioxidant enzymes. A previous report stated that ozone may be a promising new strategy in anti-cancer therapy [30]. There is evidence that the exposure of cells to frequencies with a low electromagnetic field can induce immune cell activation [31], leading to benefits for cancer patients. All in all, studies into the effects of the physical components of plasma other than ROS are needed in order to elucidate their specific roles in cancer research.

When we apply plasma to the tissue, the choice of gas and penetration of plasma are important components. Helium gas is frequently used for biomedical treatments due to the advantages of its ease of use, its high production of reactive radicals, and its low plasma temperature. ROS may penetrate cells and can induce DNA damage, which leads to the induction of apoptosis in cancer cells. Even based on the limited existing research, we can assume that plasma is able to generate paracrine effects that propagate far into the tissue. There may be unidentified side effects of long-term treatment with plasma; however, because the treatment usually lasts for 30 s at a low temperature (even after 10 min, the temperature stays at 37 °C), no significant side effects are expected.

The clinical application of CAP has been tested in many in vivo studies. After CAP was applied to a mouse glioblastoma orthotopic model and a breast cancer xenograft model, the size of the tumor decreased in plasma-treated groups, compared to non-treated groups [32,33]. In clinical practice concerning canine osteosarcoma, considering its high incidence of pulmonary metastasis, drug targeting for metastasis and examinations with thoracic radiography should be inspected [1]. As shown in Figure 5, a reduced invasion activity was observed with CAP-treated canine osteosarcoma cells, in addition to cell growth arrest and apoptosis induction. Another concern in treating canine osteosarcoma is drug resistance [34]. Interestingly, many researchers suggest CAP to overcome drug resistance in cancer. For example, CAP restored paclitaxel sensitivity to paclitaxel-resistant breast cancer cells [35]. The degree of drug uptake did not change, but the gene expression profile resulted in drug sensitivity. A similar result was reported with tamoxifen [36]. Moreover, CAP-treated glioblastoma cells showed sensitivity to the anticancer drug temozolomide [37]. Taking this into consideration, applying CAP to canine osteosarcoma as an adjuvant therapy would increase the efficacy of chemotherapy. For example, bisphosphonate is a bone-modifying agent which is used in treating human bone cancer, as it reduces skeletal complications by inhibiting bone resorption [38]. Thus, it would be worth trying the application of CAP with bisphosphonate therapy. Moreover, as mentioned earlier, hypoxia-induced angiogenesis occurs in both primary and metastatic tumors. Considering the effect of tumor microenvironments, many angiogenesis-immune interfering agents are used in human medicine [39]. Co-treating CAP with these drugs may be another option to consider.

In this study, we elucidated the anticancer effects of CAP on canine osteosarcoma cells, and we further provide the rationale for the potential use of CAP in patients where limb-sparing surgery is performed. CAP is also reported to induce immunogenic cell death [40,41]. Applying CAP on the surgical site after removing the tumor could not only kill the residual cancer cells in situ, but also suppress potential metastasis, and could activate an immune response to the residual cancer cells, considering that tumor recurrence is reported in 15–25% of the patients [34]. Furthermore, CAP has a beneficial effect on wound healing [42,43,44]. We assume that applying CAP on surgical sites could accelerate wound closure. Moreover, because postoperative infections occur frequently, in 30–50% of the patients [34], the sterilizing ability of CAP could reduce their incidence.

## 4. Materials and Methods

### 4.1. Cell Cultures and CAP Induction

The DSN canine osteosarcoma cell line (ATCC CRL-9939) was generously provided by Dr. Gwonhwa Song [45,46], and the D-17 cell line (ATCC CCL-183) was used as previously reported [47]. Both canine osteosarcoma cell lines were cultured in Dulbecco’s Modified Eagle’s Medium (DMEM; Gibco Life Technologies, Carlsbad, CA, USA) supplemented with 10% fetal bovine serum (FBS; Thermo Fisher Scientific, Waltham, MA, USA), and 1% penicillin/streptomycin (Gibco Life Technologies), maintained at 37 °C with 5% CO_2_ under humid conditions. Cells were grown on 96-, 24- or 12-well plates, with 200 μL, 1 mL or 2 mL of media, respectively. In this study, CAP (PSM Inc., Seongnam, Korea) was generated by helium gas at 300 W, using a DBD method. The distance between the nozzle and the surface of the media was fixed at 10 mm.

### 4.2. ROS Detection

A total of 10,000 cells were seeded in a 96-well plate with 200 μL of complete media and cultured overnight. The cells were stained with 10 μM of H_2_DCFDA (Sigma-Aldrich, St. Louis, MO, USA) for 30 min at 37 °C and treated with CAP. After 1 h, images of cells were taken with a Nikon Eclipse Ti fluorescence microscope (Nikon, Tokyo, Japan). For quantitative analysis of ROS, the cells were stained with 2 μg/μL of Hoechst 33342 (Sigma-Aldrich, St. Louis, MO, USA) for 5 min at 37 °C, and the fluorescence intensity was measured using the CellInsight CX7 LZR High Content Screening (HCS) Platform (Thermo Fisher Scientific, Waltham, MA, USA).

### 4.3. Western Blot Analysis

D-17 cells were cultured on a 12-well plate and treated with CAP. After 24 h, cells were lysed with RIPA Cell Lysis Buffer (1X) with EDTA (GenDEPOT, Katy, TX, USA) supplemented with 100 μM of phenylmethylsulfonyl fluoride and sodium orthovanadate, respectively. Cell lysates were separated by 10% sodium dodecyl sulfate-polyacrylamide gel electrophoresis. After transfer to a nitrocellulose membrane (GVS Filter Technology, Sanford, ME, USA), the membrane was blocked with 5% skim milk in Tris-buffered saline (TBS) with 0.05% Tween 20 (TBS-T) for 1 h, followed by incubation with an anti-phospho-Histone H2A.X (Ser139) antibody (1:1000) (2577S; Cell Signaling Technology, Danvers, MA, USA) overnight. Three washes with TBS-T were performed for 10 min each. The membrane was incubated with a secondary antibody conjugated with horseradish peroxidase (1:5000) for 1 h at room temperature, and it was washed again with TBS-T. Phospho-histone H2A.X was detected using an enhanced chemiluminescence (ECL) western blotting detection reagent (Thermo Fisher Scientific, Waltham, MA, USA). The western blot image was taken using Alliance Q9 mini (UVTEC CAMBRIDGE, Cambridge, England, UK), which was analyzed with ImageJ software (National Institutes of Health, Bethesda, MD, USA). Anti-β-actin antibody (sc-47778; Santa Cruz Biotechnology, Dallas, TX, USA) was used for loading control.

### 4.4. Cell Viability Assay

Cell viability assay was conducted using the CellTiter 96 AQ_ueous_ One Solution (Promega, WI, USA) in accordance with the manufacturer’s protocol. Briefly, 5000 cells were seeded in 200 μL of complete media in a 96-well culture plate and grown overnight. Cells were treated with CAP and incubated for 24 h. After removing 100 μL of media, 20 μL of reagent was added to each well and incubated for 2 h. Absorbance was measured at 490 nm using a spectrophotometer.

### 4.5. De novo DNA Synthesis Detection

EdU was used to measure the newly synthesized DNA. Cells were seeded in a 96-well plate and incubated overnight. After treatment with CAP and incubation for 24 h, the cells were stained with EdU using Click-iT™ Plus EdU Cell Proliferation Kit for Imaging, Alexa Fluor™ 488 dye (Invitrogen, Carlsbad, CA, USA) in accordance with the provided protocol. Fluorescence intensity was analyzed with the CX7 LZR HCS platform. All the groups (control to CAP-treated groups) were placed in the same 96-well plate, and all the exposure times were the same.

### 4.6. Apoptotic Bodies and Mitochondrial Membrane Potential Detection

Nuclear condensation and mitochondrial membrane potential were observed to detect apoptosis. Cells were fixed with 4% paraformaldehyde (Biosesang, Gyeonggido, Korea) for 10 min and stained with 10 μg/mL of DAPI (Sigma-Aldrich, St. Louis, MO, USA) for 5 min. After washing with phosphate-buffered saline (PBS) several times, nuclear condensation was observed with a fluorescence microscope. Mitochondrial membrane potential analysis by high-content screening was previously reported [48]. Briefly, cells were stained with 1 μg/mL Hoechst 33342 (Sigma-Aldrich, St. Louis, MO, USA) and 100 nM of MitoTracker Orange CMTMRos (Thermo Fisher Scientific, Waltham, MA, USA) for 15 min at 37 °C. After washing with PBS, fluorescence intensity was measured using the CX7 LZR HCS platform. All the groups (control to CAP-treated groups) were placed in the same 96-well plate, and all the exposure times were the same.

### 4.7. Detection of Apoptotic Cells Using Flow Cytometry

Cells cultured on a 12-well plate were treated with CAP and incubated for 24 h. A gate was placed on the unstained cells, eliminating the debris and clumps, and avoiding false positive and negative results. Cells were stained with Annexin V-FITC and propidium iodine using FITC Annexin V Apoptosis Detection Kit with PI (Biolend, San Diego, CA, USA) in accordance with the provided manual. Cells were analyzed using the flow cytometer CytoFLEX (Beckman Coulter, Brea, CA, USA).

### 4.8. Cell Migration Assay

Cells were grown to confluence in a 24-well plate. A scratch was made manually using a yellow pipette tip. Cells were washed with PBS several times, and the medium was replaced with 1 mL of fresh complete media. The cells were treated with CAP and incubated for 48 h. Images taken before and after CAP treatment were analyzed with ImageJ software.

### 4.9. Three-Dimensional Tumor Spheroid Invasion Assay

Tumor spheroid invasion assay was performed as reported previously [49]. Briefly, 1000 cells/200 μL were seeded in an ultra-low attachment (ULA) 96-well round bottom plate (Corning, Kennebunk, ME, USA) and incubated for 4 days. Spheroids were treated with CAP and incubated for 24 h. After removing 100 μL of media, 100 μL of Matrigel (Corning, Kennebunk, ME, USA) was added to each well, and the plate was incubated for 1 h at 37 °C. Next, 100 μL of complete media was added, and the cells were incubated for 96 h. Images were taken every 24 h and spheroid volumes were estimated with ImageJ software. The volume was calculated using the following formula: 0.5 × Length × Width^2^.

### 4.10. Statistical Analysis

Student’s *t* test was used for statistical analysis. Data were considered statistically significant when the *p* value was lower than 0.05.

## 5. Conclusions

Overall, we found that CAP generated by the DBD method induces cell growth arrest and apoptosis, with decreased migration and invasion activity in canine osteosarcoma cells. Our study opens a new era of using plasma machines in veterinary research.

## Figures and Tables

**Figure 1 ijms-21-04556-f001:**
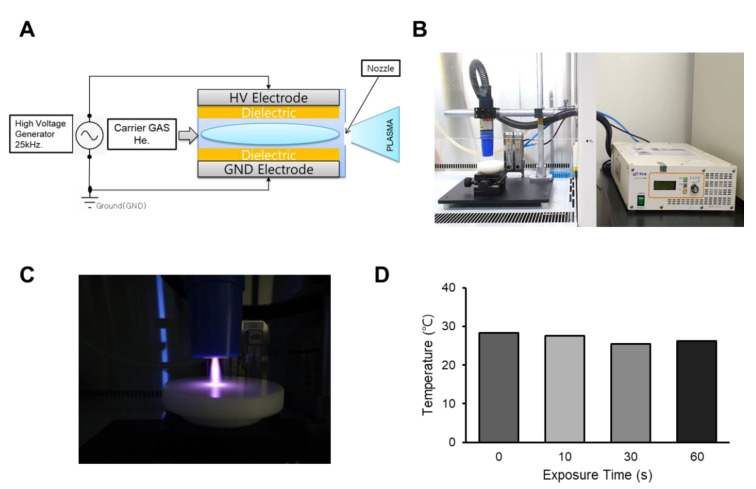
Cold atmospheric plasma (CAP) generated by dielectric-barrier discharge (DBD). (**A**) Schematic diagram of the DBD plasma device. (**B**) A photograph showing the plasma torch and its associated power supply used in this study. The plasma jet was placed on the clean bench, and all the experiments were conducted in the hood. (**C**) Representative image of cold atmospheric plasma jet using helium gas. The power was adjusted to 300 W, and the flow rate of helium (He) gas was 20 L per minute. (**D**) After CAP application to the medium, the temperature was measured at the indicated time.

**Figure 2 ijms-21-04556-f002:**
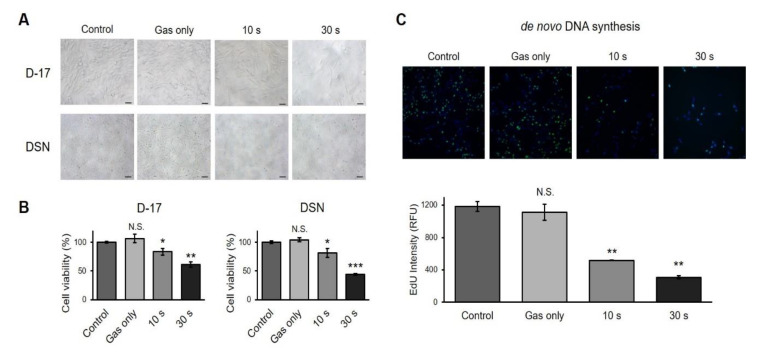
Induction of cell growth arrest by cold atmospheric plasma. (**A**) Representative images of the D-17 and DSN canine osteosarcoma cell lines treated with CAP. Scale bars, 100 μm. (**B**) Cell viability assay. D-17 and DSN cells were exposed to CAP at the indicated time. Cell viability was measured using CellTiter 96 AQ_ueous_ One Solution. Error bars represent the mean ± S.E.M. of three replicates. (**C**). Measurement of de novo DNA synthesis. D-17 cells were stained with 5-ethynyl-2′-deoxyuridine (EdU), which stains the newly synthesized DNA, and the intensity was quantified by high-content screening technology. Green, EdU; and Blue, Hoechest 33342. Error bars represent the mean ± S.E.M. of three replicates. Magnification, 100X. * *p* < 0.05, ** *p* < 0.01, *** *p* < 0.001; N.S. indicates not significant.

**Figure 3 ijms-21-04556-f003:**
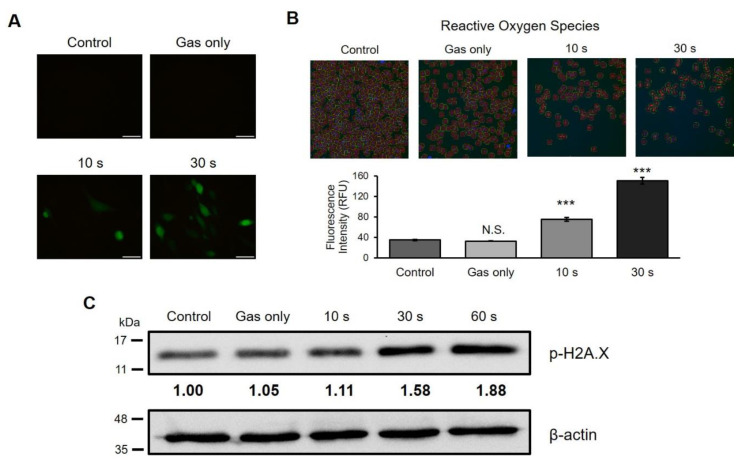
Reactive oxygen species (ROS) generation and DNA damage after exposure to cold atmospheric plasma (CAP). (**A**) Representative images of the canine osteosarcoma D-17 cell line after exposure to CAP at the indicated time. CAP-treated cells were stained with H_2_DCFDA and photographed under a fluorescence microscope. H_2_DCFDA is converted to DCF by ROS. DCF, 2′,7′-dichlorodihydrofluorescein. Scale bars, 50 μm. (**B**) Quantitative analysis of ROS using high-content screening. CAP-treated D-17 cells stained with H_2_DCFDA and Hoechst 33342 were measured by high-content screening technology. Error bars represent the mean ± S.E.M. of three replicates. Magnification, 100X. (**C**) Western blot analysis of DNA damage. Expression of phospho-histone-H2A.X was measured to assess DNA damage. This result represents two independent experiments. The intensity was normalized to β-actin. *** *p* < 0.001; N.S. indicates not significant.

**Figure 4 ijms-21-04556-f004:**
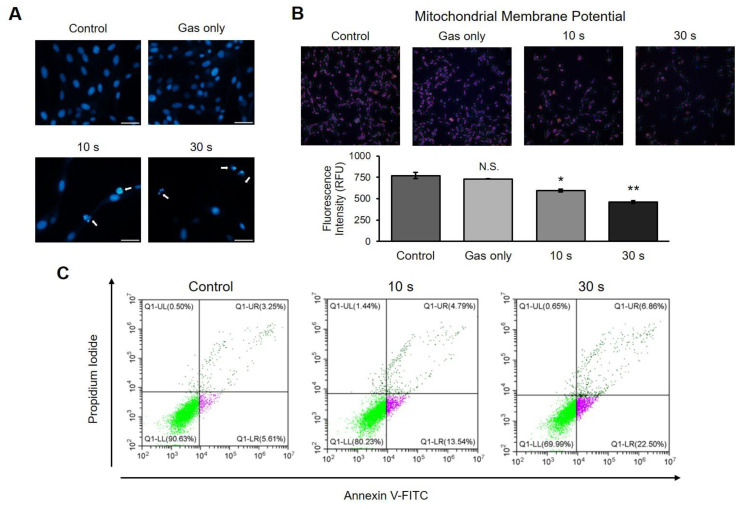
Induction of apoptosis by cold atmospheric plasma (CAP) on canine osteosarcoma cells. (**A**) Microscopy analysis of chromatin condensation. Cells were stained with 4′,6-diamidino-2-phenylindole (DAPI) and observed under a fluorescence microscope. Apoptotic nuclei were detected in CAP-treated wells. Scale bars, 50 μm. (**B**) Measurement of change in mitochondrial membrane potential (Δ ψm). The cells were stained with MitoTracker, which is a stain for the mitochondrial membrane, and the intensity was measured by high-content screening technology. CAP-treated cells showed decreased Δ ψm compared to that in control. Error bars represent the mean ± S.E.M. of three replicates. Magnification, 100X. * *p* < 0.05, ** *p* < 0.01; N.S. indicates not significant. (**C**) Detection of Annexin V-positive cells. Cells were stained with Annexin V-FITC and propidium iodide and measured using flow cytometry. CAP induced an increase in Annexin V-positive cells in a time-dependent manner.

**Figure 5 ijms-21-04556-f005:**
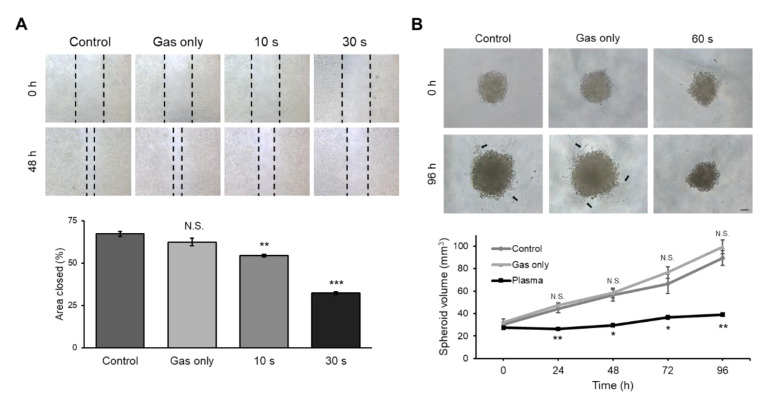
Migration and invasion activity of D-17 cells after exposure to cold atmospheric plasma (CAP). (**A**) Cell migration assay. Representative images of cells before and after exposure to CAP. Magnification, 40X. Quantitative analysis of the area refilled after 48 h. The graph represents three independent experiments. Error bars represent the mean ± S.E.M. (**B**) Three-dimensional tumor spheroid invasion assay. Representative images of spheroids before and after exposure to CAP. Invadopodia were observed in control and gas-treated group (black arrow). Scale bar, 100 μm. Volume of the control or CAP-treated spheroids was measured and is presented as a graph. Error bars represent the mean ± S.E.M. of three replicates. * *p* < 0.05, ** *p* < 0.01, *** *p* < 0.001, compared to control; N.S. indicates not significant.

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
