# Peer review of "Anticancer Effects of Cold Atmospheric Plasma in Canine Osteosarcoma Cells"

_ijms, 2020, doi:10.3390/ijms21124556_

Round 1
Reviewer 1 Report
General comments
The submitted manuscript is an interesting approach to apply CAP on canine osteosarcoma cells. The methods applied are suitable to determine apoptosis, cell growth, migratory activity and invasion of the treated cells. However, in particular the Results and the Discussion passage need thoroughly revision. One general question that remains open to me is how could cancer/a tumor be treated in vivo in a patient with this method?
Specific comments
Introduction
Some paragraphs should be rephrased as they are difficult to understand. For example:
Page 1, line 32-34: “There are distinctive metabolic pathways and different responses to many drugs in in dogs, but, compared to human cancer, new treatment approaches for canine cancer are needed.”
Material and Methods
Please give ATCC catalog numbers to identify cell lines.
DSN is an osteosarcoma cell line originated from the D17 cell line. Why did you choose to use these cells? Did you expect to receive different results compared to D17?
How many replicates were performed for each experiment?
How was the spheroid cell invasion measured/quantified (lines 292-300)?
Results
To my opinion, the first paragraph in the Results section belongs to Material and Methods (page 2, lines 64-76).
Page 8 line 213: “cellular tissue”? should be changed to ..”tissue”
Page 8. Line 216: … canine osteosarcoma cells….as it were in vitro cell culture experiments only
Fluorescence images are very small and are therefore very difficult to judge.
Discussion
This section needs to be rewritten as it is not at all specifically addressed to own results and their context with literature but consists only of general comments of CAP effects on cells.
Author Response
Response to the review of No.: ijms-822339
We would like to thank both reviewers very much for their thoughtful reading of our manuscript and constructive criticism. We have made additions to the text to address comments and concerns. The following is a recap of the reviewer comments in bold, followed by our responses.
Reviewer 1
General comments
The submitted manuscript is an interesting approach to apply CAP on canine osteosarcoma cells. The methods applied are suitable to determine apoptosis, cell growth, migratory activity and invasion of the treated cells. However, in particular the Results and the Discussion passage need thoroughly revision. One general question that remains open to me is how could cancer/a tumor be treated in vivo in a patient with this method?
Response) We thoroughly revised “Results and Discussion”. In particular, we clarified a potential therapy treatment of plasma in vivo to the discussion section.
Specific comments
Introduction
Some paragraphs should be rephrased as they are difficult to understand. For example:
Page 1, line 32-34: “There are distinctive metabolic pathways and different responses to many drugs in in dogs, but, compared to human cancer, new treatment approaches for canine cancer are needed.”
Response) For clarity, we rephrased the paragraphs to:
Since canine and human metabolic pathways and drug responses differ, there is a pressing need for new treatments regarding canine cancer.
Material and Methods
Please give ATCC catalog numbers to identify cell lines.
Response) As per requested, we added the catalogue numbers for those cells.
D-17 (ATCC® CCL-183™) and DSN (ATCC® CRL-9939™)
DSN is an osteosarcoma cell line originated from the D17 cell line. Why did you choose to use these cells? Did you expect to receive different results compared to D17?
Response) Although DSN is derived from D-17, it may show different characteristics from its original cell line. There are some reports that DSN is more sensitive to anticancer drugs and natural compounds [1, 2]. We also showed in Fig. 2 that DSN was more vulnerable to cold atmospheric plasma. In addition, these are the only canine osteosarcoma cells that we could purchase from ATCC.
How many replicates were performed for each experiment?
Response) The replicates are indicated in the figure legend.
How was the spheroid cell invasion measured/quantified (lines 292-300)?
Response) Spheroid cell invasion assay was shown in Fig. 4B. We did not quantify the invasion activity, but rather presented a picture displaying the outgrowth from the spheroids (indicated by arrows). However, the volume of spheroids was also measured as described in the Methods section and presents in Fig. 4B as a graph. We rewrote this part for clarity.
Results
To my opinion, the first paragraph in the Results section belongs to Material and Methods (page 2, lines 64-76).
Response) We strongly believe that this paragraph should be in the Results section. Although it may seem that we are explaining the method we took to come to our conclusions, this machine was made for the first time and we have characterized the physical properties of this machine. Like most plasma-related manuscripts, previous publishers usually have characterized their machines in the first section of results.
Page 8 line 213: “cellular tissue”? should be changed to ..”tissue”
Response) We changed it accordingly.
Page 8. Line 216: … canine osteosarcoma cells….as it were in vitro cell culture experiments only
Response) We changed it accordingly.
Fluorescence images are very small and are therefore very difficult to judge.
Response) We reorganized figures by enlarging fluorescence images.
Discussion
This section needs to be rewritten as it is not at all specifically addressed to own results and their context with literature but consists only of general comments of CAP effects on cells.
Response) We have thoroughly revised the discussion section.
Reviewer 2 Report
Jaehak Lee and colleagues report about novel pre-clinical approaches to sarcomas in dogs. In details, they propose cold atmospheric plasma as an anti-oncogenic tool that can represent a promising extension to existing applications. The topic is of some interest. Nonetheless, few major concerns prevent the publication in IJMS in this format and should be addressed.
Major point:
Figure 2 c, 3 a,b, 4 a,b: can the authors comment on the exposure time of the immune-fluorescence images? Were all the same and normalized for the different conditions? Those are important information and should be mentioned.
Figure 3 c: for all Western blot figures, densitometry readings/intensity ratio of each band should be included; the whole Western blot showing all bands and molecular weight markers should be included in the Supplementary Materials
Figure 4 c: FACS plot: did the authors used unstained/isotype control? If this is the case, it should be mention. Otherwise, an explanation should be provided.
Migration and invasion assay (Fig 5): did the authors check on cell viability/proliferation rate in order to discriminate the effect related to epithelial-mesenchymal transition (EMT) phenotype influence independent from toxic/antiproliferative effects
Statistical analysis: the authors state that they employed t student for analysis. This is fine as long as the analyzed data are normally distributed, otherwise a non-parametric test should be employed. It would be useful to receive the authors feedback on this topic.
General comments: This reviewer’s concern is triggered by the translational potential of the presented results. The clinical and pre-clinical background by which such an approach seems to be prompted does not look strong enough, at least in the way is presented. Does clinical application in the field of oncology exist and substantiate the applicability on real-life scenario of cold atmospheric plasma? A suggestion might be to expand the introduction/discussion section in light of some therapeutic windows the authors indirectly allude to.
Namely, in the introduction and discussion section (ref 1, 8, 20, 21) the authors briefly mention the potential implication on tumour microenvironment and bone disease. I miss some important implication of cold atmospheric plasma while considering oxygen pressure and its implication: is required for our organs to function properly. Conversely, insufficient oxygen supply (hypoxia) is a prominent feature in various pathological processes, including tumour development and metastasis. During different stages of tumour development, these insights may significantly impact the authors' frame of thinking, whilst this oxygen sensor is also essential during bone mineralization and normalization of the endothelial barrier in the bone (marrow) after stress in both solid and haematological malignancies. Because of these intimate interactions between stressors, microenvironment and drug sensitivity in osteotropic cancers, the potential of combining the proposed tool (CAP) with angiogenesis-immune interfering agents and bone-modifying agents, can be also discussed, since several examples have been published (PMID: 31470608, PMID: 30046358; PMID: 27474171) and can increase the biological and translational landscape of the presented results also in human disease landscape, while increasing the interest for a broad readership in oncology.
Author Response
Response to the review of No.: ijms-822339
We would like to thank both reviewers very much for their thoughtful reading of our manuscript and constructive criticism. We have made additions to the text to address comments and concerns. The following is a recap of the reviewer comments in bold, followed by our responses.
Reviewer 2
Comments and Suggestions for Authors
Jaehak Lee and colleagues report about novel pre-clinical approaches to sarcomas in dogs. In details, they propose cold atmospheric plasma as an anti-oncogenic tool that can represent a promising extension to existing applications. The topic is of some interest. Nonetheless, few major concerns prevent the publication in IJMS in this format and should be addressed.
Major point:
Figure 2 c, 3 a,b, 4 a,b: can the authors comment on the exposure time of the immune-fluorescence images? Were all the same and normalized for the different conditions? Those are important information and should be mentioned.
Response) We do realize the reviewer’s concern about intentionally manipulating exposure time to get an expected result (e.g. long exposure time for control and short exposure time for CAP-treated group). Fluorescence images were taken and quantified using High-content screening (HCS). HCS is an image-based analysis technique/instrument, which takes fluorescence images and measures the intensity simultaneously in one attempt. All the groups (control to CAP-treated groups) were placed in the same 96-well plate, and all the exposure times were the same. The exposure time in detail is as followed:
|
De novo DNA synthesis (Fig.2c) |
|||||||||||
|
|
Hoechest 33342 |
EdU |
|||||||||
|
Exposure time (s) |
0.25 |
0.1 |
|||||||||
Mitochondrial Membrane Potential (Fig.4b) |
|||||||||||
|
|
Hoechest 33342 |
Mitotracker |
|||||||||
|
Exposure time (s) |
0.1 |
0.01 |
|||||||||
The exposure time was chosen by checking the dynamic range of the fluorescence to avoid over-saturation before starting the experiment. For Hoechest 33342 (nucleus staining), the exposure time was decided when the amount of saturation was approximately 20-25%, and the target fluorescence for approximately 10-16%. Additionally, images in Fig. 3a were taken for 0.5 s and images in Fig 4a. for 0.3 s.
We have decided not to include the various times (table shown above) with the paper, but rather mention that with each method, we used the same times.
Figure 3 c: for all Western blot figures, densitometry readings/intensity ratio of each band should be included; the whole Western blot showing all bands and molecular weight markers should be included in the Supplementary Materials
Response) For the western blot data, band intensity ratio normalized to loading control is indicated, and uncropped images are below. We will add these to the supplementary materials.
phospho-H2A.X
phospho-H2A.X with markers
beta-actin
Figure 4 c: FACS plot: did the authors used unstained/isotype control? If this is the case, it should be mention. Otherwise, an explanation should be provided.
Response) Yes, we used unstained cells to form a gate to single cell population prior to detecting fluorescence positive cells. From unstained cells, cell population was gated and final data were analyzed. For clarity, we added this in the method section.
“A gate was placed on the unstained cells eliminating the debris and avoiding false positive and negative results.”
Migration and invasion assay (Fig 5): did the authors check on cell viability/proliferation rate in order to discriminate the effect related to epithelial-mesenchymal transition (EMT) phenotype influence independent from toxic/antiproliferative effects
Response) As shown in Fig 2B, CAP affects cell viability. Therefore, cell migration assay may be due to anti-proliferative activity of plasma. However, outreach growth in 3D spheroid assay has nothing to do with the anti-proliferative activity of plasma. Further analysis may be needed for the conclusion, but we believe that CAP affects anti-metastatic activity.
Statistical analysis: the authors state that they employed t student for analysis. This is fine as long as the analyzed data are normally distributed, otherwise a non-parametric test should be employed. It would be useful to receive the authors feedback on this topic.
Response) We checked whether the data are normally distributed by kolmogorov-smirnov test. It is normally distributed.
General comments: This reviewer’s concern is triggered by the translational potential of the presented results. The clinical and pre-clinical background by which such an approach seems to be prompted does not look strong enough, at least in the way is presented. Does clinical application in the field of oncology exist and substantiate the applicability on real-life scenario of cold atmospheric plasma? A suggestion might be to expand the introduction/discussion section in light of some therapeutic windows the authors indirectly allude to.
Response) As commented, we intensively revised the discussion section including a new paragraph related to clinical application.
Namely, in the introduction and discussion section (ref 1, 8, 20, 21) the authors briefly mention the potential implication on tumour microenvironment and bone disease. I miss some important implication of cold atmospheric plasma while considering oxygen pressure and its implication: is required for our organs to function properly. Conversely, insufficient oxygen supply (hypoxia) is a prominent feature in various pathological processes, including tumour development and metastasis. During different stages of tumour development, these insights may significantly impact the authors' frame of thinking, whilst this oxygen sensor is also essential during bone mineralization and normalization of the endothelial barrier in the bone (marrow) after stress in both solid and haematological malignancies. Because of these intimate interactions between stressors, microenvironment and drug sensitivity in osteotropic cancers, the potential of combining the proposed tool (CAP) with angiogenesis-immune interfering agents and bone-modifying agents, can be also discussed, since several examples have been published (PMID: 31470608, PMID: 30046358; PMID: 27474171) and can increase the biological and translational landscape of the presented results also in human disease landscape, while increasing the interest for a broad readership in oncology.
Response) We really appreciate the reviewer’s comment and have made revisions to our paper.

Round 2
Reviewer 2 Report
The authors have clarified several of the questions I raised in my previous review. Most of the major problems have been addressed by this revision.
No further comments from this reviewer.